# An Experimental Approach to Study the Effects of Realistic Environmental Mixture of Linuron and Propamocarb on Zebrafish Synaptogenesis

**DOI:** 10.3390/ijerph18094664

**Published:** 2021-04-27

**Authors:** Giulia Caioni, Carmine Merola, Monia Perugini, Michele d’Angelo, Anna Maria Cimini, Michele Amorena, Elisabetta Benedetti

**Affiliations:** 1Department of Life, Health and Environmental Sciences, University of L’Aquila, 67100 L’Aquila, Italy; giulia.caioni@guest.univaq.it (G.C.); michele.dangelo@univaq.it (M.d.); annamaria.cimini@univaq.it (A.M.C.); elisabetta.benedetti@univaq.it (E.B.); 2Faculty of Bioscience and Technology for Food, Agriculture and Environment, University of Teramo, 64100 Teramo, Italy; cmerola@unite.it (C.M.); mamorena@unite.it (M.A.)

**Keywords:** pesticides, mixtures, zebrafish, synaptogenesis, sublethal effects

## Abstract

The reasons behind the extensive use of pesticides include the need to destroy vector organisms and promote agricultural production in order to sustain population growth. Exposure to pesticides is principally occupational, even if their persistence in soil, surface water and food brings the risk closer to the general population, hence the demand for risk assessment, since these compounds exist not only as individual chemicals but also in form of mixtures. In light of this, zebrafish represents a suitable model for the evaluation of toxicological effects. Here, zebrafish embryos were exposed for 96 h post fertilization (hpf) to sublethal concentrations (350 µg/L) of linuron and propamocarb, used separately and then combined in a single solution. We investigated the effects on morphological traits and the expression of genes known to be implicated in synaptogenesis (*neurexin1a* and *neuroligin3b*). We observed alterations in some phenotypic parameters, such as head width and interocular distance, that showed a significant reduction (*p* < 0.05) for the mixture treatment. After individual exposure, the analysis of gene expression showed an imbalance at the synaptic level, which was partially recovered by the simultaneous administration of linuron and propamocarb. This preliminary study demonstrates that the combined substances were responsible for some unpredictable effects, diverging from the effect observed after single exposure. Thus, it is clear that risk assessment should be performed not only on single pesticides but also on their mixtures, the toxicological dynamics of which can be totally unpredictable.

## 1. Introduction

Environmental pollutants are recognized as a major concern for public health, and are responsible for various neurological disorders [1]. Typically, several thousand compounds are detectable in environmental samples, including pesticides, pharmaceuticals and heavy metals [2]. Knowledge regarding the neurotoxic potential of environmental contaminant mixtures is very limited, since the assessment of neurotoxicity is currently mostly focused on human exposure to individual chemicals. It is well-documented that “mixture effects” can be greater than the effects triggered by the most potent single chemical in a mixture due to their additive or, in some cases, even synergistic effects [3]. In wildlife, as in humans, early life stages are susceptible to toxicant insults, and developmental neurotoxicity represents an issue of major concern. The developing brain is uniquely vulnerable to toxic chemical exposures. During these sensitive life stages, chemicals can cause permanent brain injury at low levels of exposure that would have little or no adverse effect in adult organisms [4]. Despite this particular concern, the data regarding the neurotoxicity of chemical mixtures are scarce. A few chemicals were tested for potential human developmental neurotoxicity following respective OECD or US-EPA guidelines [5]. Therefore, there is a demand for time- and cost-efficient testing methods to evaluate many chemicals for developmental neurotoxicity in both single and mixed exposure [2]. Zebrafish early life stages offer several advantages for studying developmental neurotoxicity [6,7]. Within the risk assessment and risk management of neurotoxic substances, pesticides are a chemical class of special interest. They are substances or preparations that repel, destroy or control pests [8]. As a class of compounds, pesticides are, in practice, composed of many subclasses that are generally divided based on their target pests (e.g., insecticides, herbicides, fungicides, rodenticides) [9]. Ideally, pesticides injurious action would be highly specific for undesirable species; however, their molecular targets are often shared between pest and non-target species, including humans [8,10]. This concept is particularly true for insecticides (e.g., organochlorine, organophosphate and pyrethroid), which kill insects by disrupting their nervous systems and exerting neurotoxic effects on humans and other species [8]. Although herbicides and fungicides theoretically should not have shared targets with mammals, they might also have neurotoxic potential [1]. Thus, pesticide application could have substantial impacts on human health, especially for occupational health exposures. Therefore, governmental actions, educational programs and training for farmers on the safe use of pesticides are required [11,12].

The present study aims to investigate the effects of sublethal concentrations of two largely used pesticides (linuron (LIN), an herbicide, and propamocarb (PM), a fungicide) on zebrafish early life stages, after single and combined exposure. Both LIN and PM have recently been investigated in zebrafish early life stages, identifying their ability to impair neurotransmitter biosynthesis and affect the transcription levels of several neurotoxicity-related genes [13,14]. Regarding LIN and PM neurotoxicity, their ability to interfere with neurotransmitter production, release and uptake (e.g., gamma-aminobutyric acid (GABA), dopamine (DOPA), acetylcholine (ACh)), synapse formation, glial cells and neuronal differentiation is well documented [13,14]. However, there is still a lack of knowledge regarding these two pesticides’ mixture effects on zebrafish early life stages, and thus studies on possible effects of the realistic environmental mixture of LIN and PM on zebrafish synaptogenesis are necessary.

In this study, we conducted phenotypic analysis and gene expression studies through real-time PCR. Particularly, we focused our study on the head traits of developing zebrafish, and on the expression levels of two important genes involved in brain development and function, *neurexin1a* and *neuroligin3b* (*nrxn1a* and *nlgn3b*). Both *nrxn1a* and *nlgn3b* are known to be involved in synapse maturation and organization in vertebrates [15]. Neurexins are predominantly pre-synaptic cell adhesion molecules. They can induce pre-synaptic differentiation by interacting with neuroligins. There are three neurexin genes (*nrxn1*, *2* and *3*), each of which encodes two major variants (alpha and beta) [16]. Disruption of the *nrxn1* gene has been associated with autism spectrum disorders and schizophrenia in humans [17]. Two orthologs of *nrxn* have been identified in zebrafish (*nrxn1a* and *1b*) with over 70% shared identity to the human proteins [18]. An expression analysis showed that all three *nrxn* genes are expressed during zebrafish embryonic development and some specific isoforms of *nrxn1a* are expressed at different stages of development [16]. During the larval stage, the synaptogenesis increases progressively in all the major regions of the central nervous system (CNS) and in the myotome, highlighting the potential harmful effects of chemical exposure in this specific temporal window. Neurexins perturbation resulted in an attenuation of neurotransmitter release and in an increase of synapse elimination [19]. Neuroligins are post-synaptic cell adhesion membrane proteins that trans-synaptically interact with pre-synaptic neurexins [20]. Rodents and humans have four neuroligins (NLGNs *1–4):* NLGN1 is selectively localized in excitatory synapses, NLGN2 in inhibitory synapses and NLGN4 in glycinergic synapses. NLGN3 is present in both excitatory and inhibitory synapses, making NLGN3 of particular interest [21]. Mutation and deletions in *Nlgn* genes have been linked to autism spectrum disorders and schizophrenia [22]. In zebrafish, *nlgn2*, *3* and *4* genes are duplicated; only the *nlgn1* gene seems to be present in a single copy. In zebrafish embryos, *nlgn3b* is not expressed until the 8-somite stage. Notably, starting from 24 hpf its expression increases, and at 48 hpf, *nlgn3b* mRNA is detectable in the posterior telencephalon, dorsal and ventral diencephalon and the ventral portion of the rhombencephalon. Neuroligin overexpression causes large increases in synaptogenesis, and its knockdown generally induces dramatic loss of synapses, while conditional genetic deletions mostly induce impairments in synaptic function with either minor or no changes in synapse numbers [23]. Together, the neurexin–neuroligin complex contributes to the formation of a specialized area required for the correct synaptic transmission [21]. Moreover, these proteins are found to affect the brain volume of mice [24]. In particular, *Nlgn3*-deficient mice showed a decrease of brain volume, which is consistent with abnormalities also identified in autistic children [25]. Brain morphology can also be influenced by *NRXN1* levels, leading to the typical phenotypes of schizophrenia, mental retardation and autism [26].

Since there is evidence of the pesticides’ involvement in altering neural connections, our objective was to develop a series of methodologies that could better investigate the effects mediated by the LIN and PM mixture, using zebrafish as the in vivo model. This research represents a preliminary approach, which tries to cover the gap of information regarding pesticide mixture effects on aquatic organisms, and in the future it could be very interesting to investigate the joint toxic action of mixtures of multiple compounds at environmental concentrations.

## 2. Materials and Methods

### 2.1. Chemicals

LIN (CAS number 350-55-2) and PM (CAS number 24579-73-5), as well as formalin 37%, isopropanol absolute and ethanol absolute were purchased from Merck Life Science (Milano, Italy). Dilution water (DW) was prepared according to OECD TG 203, Annex 2 (OECD, 1992).

### 2.2. Zebrafish Maintenance and Egg Collection

Adult zebrafish (wild type AB strain) were bred in a University of Teramo facility (code 041TE294) and raised in 3.5 L ZebTec tanks (Tecniplast S.p.a., Buguggiate, Italy) in a recirculating aquatic system. The system conditions were the following: pH at 7 ± 0.2, conductivity at 500 ± 100 μS cm^−1^ and dissolved O_2_ at 6.1 mg/L. Temperature was maintained at 28 °C and the fish were kept under a constant artificial dark/light cycle of 10/14 h. Permanent flow-through conditions guaranteed the following values: ammonia 0.02 mg/L, nitrite 0.02 mg/L, nitrate 21.3 mg/L. The zebrafish were fed twice a day with live food (*Artemia salina*) and supplemented with ZEBRAFEED 400–600 (Sparos, Olhão, Portugal). The afternoon before spawning, ten groups of females and males (1:1) were introduced into 1.7 L breeding tanks (beach style design, Tecniplast S.p.a.). Immediately after spawning (initiated by morning light), fertilized eggs were collected with a sieve and rinsed thoroughly with deionized water and DW. Newly fertilized eggs were collected immediately after spawning and placed in groups of approximately 100 per Petri dish, within a light- and temperature-controlled incubator until 2–3 hpf. Non-fertilized eggs and embryos with injuries were eliminated.

### 2.3. Zebrafish Embryos Exposure

Stock solutions of each compound covering the test concentration were prepared in DW and stored at 4 °C. LIN and PM were tested at a concentration of 350 µg/L, in single and combined exposure. Toxicological concentrations were chosen in a range of levels reported to have sublethal effects on developing zebrafish [13,14]. At 2–3 hpf, embryos were examined under a dissecting microscope, and those embryos that had developed normally were selected for subsequent experiments. Afterward, embryos (4–16-cell stage) were transferred to crystallizing dishes (diameter 115 mm, capacity 1000 mL) with 75 embryos in 150 mL of test solution containing the respective test substance. To prevent evaporation, crystallizing dishes were covered with self-adhesive transparent foil (SealPlate by EXCEL Scientific, Dunn, Asbach, Germany). Testing solutions were changed every 24 h to maintain suitable chemical concentrations and water quality. Embryos were exposed for 96 hpf in the incubator at 26 ± 1 °C and photoperiod (14 h light:10 dark) conditions. Three replicates for each concentration were used. Negative control (Ctr), embryos in DW were also tested.

### 2.4. Morphometric Measurements

Zebrafish at 96 hpf were collected and fixed overnight at 4 °C in a phosphate-buffered saline 1× solution (PBS) with 4% formalin. Subsequently they were washed with 1× PBS and mounted on a microscope slide with 1% agar solution in bi-distilled water. A Leica S8 Apo stereomicroscope equipped with an EC3 camera (Leica Microsystem) was used to observe and acquire images. The following parameters were measured using ImageJ software: body length (BL), eye length (EL), eye width (EW), interocular distance (IOD), head length (HL), head width (HW). Thirty larvae for each condition were examined.

### 2.5. RT-PCR

Total RNA was isolated from a pool of 50 zebrafish larvae for each condition and for three replicates with Trizol^TM^ Reagent (Invitrogen, Carlsbad, CA, USA). Nucleic acid purity and RNA concentration were determined by NanoDrop^TM^ 2000 Spectrophotometer (Thermo Fisher Scientific, Waltham, MA, USA) and Qubit 2.0 fluorometer (Invitrogen, CA, USA), respectively. cDNA was synthesized using 5X All-In-One RT MasterMix (Applied Biological Materials Inc., Richmond, BC, Canada) according to the manufacturer’s instructions. RT-PCRs were performed with Applied Biosystem 7300 Real-Time PCR system, using TaqMan Universal PCR Master Mix (Thermo Fisher Scientific, Vilnius, Lithuania) and gene-specific probe (Biorad) for *nrxn1a* (qDreCEP0050779) and *nlgn3b* (qDreCIP0030702). The results were normalized using TATA-box binding protein (tbp) (qDreCIP0036647) as a reference gene. We used the 2^−ΔΔCt^ method to calculate the expression levels.

### 2.6. Statistical Analysis

For morphometric analysis and RT-PCR, results were expressed as the mean ± Standard error of the mean (SEM). Statistical analyses were performed using GraphPad Prism 7 software (GraphPad software). One-way ANOVA was used to determine the statistical difference between exposed and control groups. Differences were considered statistically significant if *p*-values were less than 0.05.

## 3. Results

### 3.1. Morphometric Analyses

Morphometric analyses showed no statistically significant effect on BL, HL or EL compared to control, after single and combined exposure to sublethal LIN and PM concentrations (Figure 1). A significant (*p* < 0.05) decrease in HW was observed only for the mixture treatment, while the zebrafish larvae treated with both PM and the mixture treatment also showed a significant reduction (*p* < 0.05 and *p* < 0.005 respectively) in the IOD (Figure 1).

### 3.2. RT-PCR

To determine the effects of LIN, PM and MIX on synaptogenesis, we investigated the mRNA expression levels of *nrx1a* and *nlgn3b*. *nrx1a* showed a significant increase of expression in larvae exposed to LIN while *nlgn3b* showed a general decrease in those exposed to PM (*p* < 0.05). The exposure to the pesticide mixture did not show any significant difference for mRNA expression levels, since *nlg3b* and *nrxn1a* levels were comparable to the control larvae (Figure 2). This result shows the different effect of these pesticides in their combined formulation.

## 4. Discussion

Developmental exposure to several pesticides, including organochlorine, organophosphate and pyrethroid, appears to impact the expression of neuronal targets critical to synaptic function [27]. Synaptic alterations following pesticide exposure appear to be relatively complex and are not selective for a particular neural circuit or brain region. Notably, synaptic dysfunction is known to be associated with neurologic and psychiatric disorders, as well as more subtle cognitive, psychomotor and sensory defects [27]. Although chemicals have always been considered on an individual basis, they usually exist as mixtures in aquatic and terrestrial ecosystems. Consequently, organisms are often simultaneously exposed to a wide variety of pesticides, highlighting the need to deepen toxicological information on the joint effects of pesticide mixtures.

LIN is a systemic and selective herbicide from the urea family used for pre- and post-emergence control of annual grass and broad-leaved weeds in cultures of several kinds of cereal, fruit and vegetables [28]. Its mechanism of action is related to the inhibition of photosynthesis by disrupting photosystem II and blocking electron transport, leading to the production of a range of oxidants and the rapid destruction of plant cells [29]. Although LIN exhibits low to medium mobility in soil, it has been found in surface waters at concentrations ranging from nanograms per liter (e.g., 2.5 ng/L in Alqueva reservoir, Portugal) to micrograms per liter (4.42 μg/L in a Florida stream) [28,30,31]. LIN has shown endocrine disruption properties with an anti-androgenic mode of action in mammals and aquatic organisms [32,33]. Meanwhile, PM is a high-efficiency, broad-spectrum, systemic carbamate fungicide that acts primarily by inhibiting the biosynthesis of phospholipids and fatty acids in the cell membrane components of the oomycetes, thereby inhibiting mycelial growth, sporangia formation and germination [14]. As a widely used fungicide to protect cucumbers and other plants from downy mildew, PM has been detected in the environment, with soil surface concentrations reaching 0.134 mg/kg [34]. Chronic and acute exposure at a wide range of PM concentrations has caused metabolic disorders and gut microbiota dysbiosis in male adult mice and zebrafish [34,35,36].

The morphometric analysis (Figure 1) showed that LIN did not affect the morphometric measures, while the PM—and above all, the combined LIN-PM treatment—influenced the HW and IOD of zebrafish larvae, highlighting the ability of these chemicals to affect the craniofacial features of exposed zebrafish. IOD was also reduced following exposure to endocrine disruptors able to negatively impact the proper development of the brain structure of developing zebrafish larvae (e.g., bisphenol A and PFOS) [37]. Combined exposure of LIN and PM exerted more pronounced and significant effects on the reduction of head width compared to single pesticide exposure. Epidemiological studies have shown that prenatal exposure to pesticides, including chlorpyrifos and non-organophosphate household pesticides, has been linked to reduced head circumference in human neonates [38,39]. Our results showed that LIN and PM could interact with each other, affecting the IOD, revealing a greater than expected threat for human health than the individual pesticides present alone.

Recently, LIN exposure at concentrations as low as 1.25 μM has been found to negatively impact larval activity in the Visual Motor Response test, causing hypoactivity in 7-day-old zebrafish larvae. Moreover, Gad1b, the rate-limiting enzyme in GABA synthesis, decreased in transcript abundance with increasing LIN concentration. A similar trend was also observed for Th1, the rate-limiting enzyme involved in dopamine production [13]. Dopamine levels and dopamine-related genes were also reduced in zebrafish larvae exposed to 1000 μg/L of PM [14]. Zebrafish larvae treated with 100 μg/L and 1000 μg/L of PM also showed reduced activity of acetylcholinesterase (AChE) and decreased mRNA level of *ache* [14]. Moreover, the levels of *elavl3, shha*, *syn2a* and *neurog1* mRNA in the 100 and 1000 μg/L PM-treated groups decreased significantly, and the mRNA levels of *nestin*, *gap43* and *gfap* decreased significantly in the 1000 μg/L PM-treated groups [14]. According to the results of the study reported above, LIN and PM affected the nervous system of zebrafish at various molecular endpoints, including those involved in neurotransmitter production and release, synapse formation, glial cells and neuronal differentiation.

Dysregulation of genes involved in transcription and translation could lead to altered numbers of neurons or glial cells in the adult brain, altered distribution of cell types in different regions, and multiple changes based on alterations in the translation of cell-specific proteins that may only have a limited window of time to function appropriately.

The RT-PCR results showed a different modulation by single and combined treatments on the expression levels of *nrxn1a* and *nlgn3b*. Regarding *nrxn1a*, LIN induces an increase in its mRNA expression level (*p* < 0.05), suggesting a modification in the GABAergic synaptic communication. Overexpression of neurexins seems not to be related to an increase of synapse number, but with the suppression of GABAergic transmissions [40] in which neurexins actively participated in the regulation of the excitatory/inhibitory balance in the brain [40]. Thus our data are in line with the decrease observed in *gad1b* after LIN treatment in [13]. Likewise, *nlgn3b* is an adhesion protein of glutamatergic and GABAergic synapses, and it is significantly reduced by the PM treatments (*p* < 0.05), inducing impairment in synaptogenesis. In fact, a study on *Nlgn3*-deficient mice revealed that the elimination of this gene could lead to several symptoms linked to autism spectrum disorder, including alterations in social memory [24]; this result is in line with previously published results in the literature.

Exposure to the single pesticide seems to promote an imbalance at the synaptic level towards excitatory function, posing a concern regarding the possible implications for neurological and cognitive activities.

The combined treatment produced opposite effects compared to the single exposure, nearly bringing *nrxn1/nlgn3b* levels back to the unexposed condition. This result indicates a change in LIN and PM activity when these compounds are mixed. Thus, a deep analysis is necessary to understand how the mixture could act on *nrx1a* and *nlng3b* expression. Nevertheless, the smaller HW of mixture-treated larvae indicates an effect on the head development resulting from this kind of treatment.

## 5. Conclusions

The effects of exposure to single pesticides (LIN and PM) did not show changes in the morphological traits, except for HW and IOD parameters. In particular, the HW became increasingly smaller when moving from a single exposure to the mixed exposure, indicating a major effect of the mixture on head morphology. Moreover, the values obtained from RT-PCR show the different action of these pesticides on the organization and function of synapses. Thus, changes in the expression of the selected genes suggest a target in the neurodevelopmental toxicity potential for these pesticides. Toxicodynamic interactions are difficult to examine, but it is clear that exposure to both pesticides determines an unexpected response. These effects may be related to the presence of independent mechanisms of action, and a deeper examination could give a more exhaustive explanation of this. Furthermore, to have a complete overview, an examination of other genes involved in both synapse maturation and on cell-adhesion-related processes is required, along with a study of behavioral manifestations attributable to changes in the neuronal network.

## Figures and Tables

**Figure 1 ijerph-18-04664-f001:**
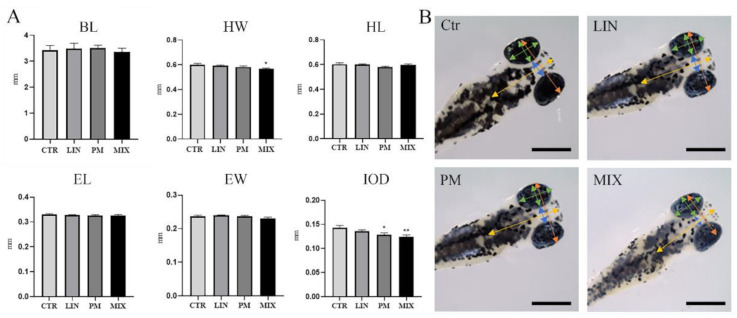
Morphometric measures of zebrafish larvae exposed to LIN and PM as single pesticides (LIN, PM) and mixtures (MIX) for 96 hpf (*n* = 50/condition). (**A**) The morphological traits measured were body length (BL), head width (HW), head length (HL), eye length (EL), eye width (EW), interocular distance (IOD). All measures are expressed in millimeters. The results are expressed as mean ± SEM. * *p* < 0.05, ** *p* < 0.005 vs. Ctr. (**B**) Representative pictures of zebrafish larvae for each condition and description of morphological traits measured (scale bar: 0.3 mm).

**Figure 2 ijerph-18-04664-f002:**
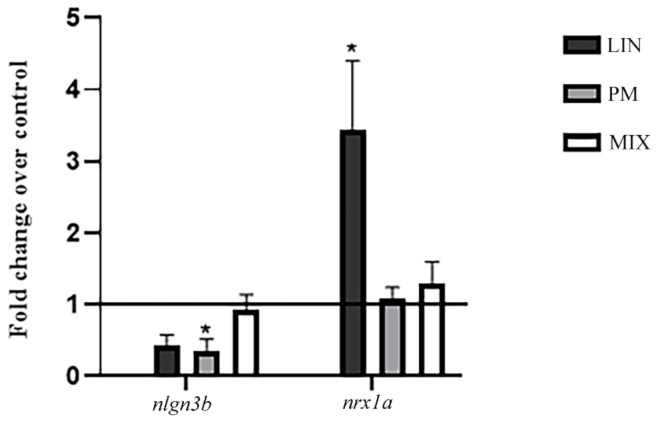
mRNA expression levels of *nlgn3b* and *nrxn1a* after exposure of zebrafish larvae to LIN, PM and MIX for 96 hpf. The results are expressed as mean ± SEM. * *p* < 0.05 vs. Ctr.

## Data Availability

Not applicable.

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
