# Peer review of "An Experimental Approach to Study the Effects of Realistic Environmental Mixture of Linuron and Propamocarb on Zebrafish Synaptogenesis"

_ijerph, 2021, doi:10.3390/ijerph18094664_

Round 1
Reviewer 1 Report
The theme is very pertinent and is and it is a welcome work to evaluate the toxicity of the mixture of two substances, widely used in agriculture. Overall, the Ms. is well written, but considerer some suggestions and concerns about your work.
Important and more specific points:
- Why you did not use the same molarity for both substances? And different mixtures proportions?
- Line 15 – “especially occupational” – considerer substitute especially by “principally”
- Please introduce quantitative data in the abstract, to reinforce some statements
- Line 115 – correct cm−1 and O2 (superscript)
- “LIN and PM were tested at concentration of 350 μg/L” – each? Even in the mixture?
- Line 133 – consistently use L for liter (check throughout the text), please see “capacity 1000 ml)” to “… 1000 mL)” and 150 ml
- Line 159 – explained the SEM acronym
- Line 182 – remove comma in “(p<0.05),.”
- Gene symbols must be in italic, and additionally, I think that in Danio, gene symbols have all letters in lowercase (please verify).
- As stated in discussion section, the environmental concentrations of LIN and PM (in soil, surface waters) is far below the tested concentration. Could the risk being overestimated?
Author Response
We have modified and highligthed in yellow the changes that we have done following the Reviewer suggestions.
- Why you did not use the same molarity for both substances? And different mixtures proportions?
The Reviewer is correct because often chemist use the molarity to calculate the levels of exposure. However, the drawback of using molarity is that volume is a temperature-dependent quantity. As temperature changes, density changes, which affects volume. Then we prefer to use the concentration of solutions that do not involve volume and are temperature independent.
- Line 15 – “especially occupational” – considerer substitute especially by “principally”.
Done
- Please introduce quantitative data in the abstract, to reinforce some statements.
As suggested by the Reviewer we have added in the abstract the p value for the phenotypic parameters.
- Line 115 – correct cm−1 and O2 (superscript).
Done
- “LIN and PM were tested at concentration of 350 μg/L” – each? Even in the mixture?
The Reviewer has well understood, the concentration for both, LIN and PM in single use and in mixture was of 350 μg/L
- Line 133 – consistently use L for liter (check throughout the text), please see “capacity 1000 ml)” to “… 1000 mL)” and 150 ml.
Done
- Line 159 – explained the SEM acronym.
As suggested by the Reviewer we have written in full the acronym and moved it in the brackets.
- Line 182 – remove comma in “(p<0.05),.”
Done
- Gene symbols must be in italic, and additionally, I think that in Danio, gene symbols have all letters in lowercase (please verify).
Thank you for the comment. The Reviewer is right. We modified gene symbols using zebrafish and mammalian naming conventions:
species / gene / protein
zebrafish /nrxn1a / Nrxn1a
human / NRXN1a/ NRXN1a
mouse / Nrxn1a / NRXN1a
- As stated in discussion section, the environmental concentrations of LIN and PM (in soil, surface waters) is far below the tested concentration. Could the risk being overestimated?
The Reviewer is right because although herbicides are widely used in the agriculture, the concentrations detected in freshwaters are generally in the range of pg to μg, but in some cases and specific countries, herbicide concentrations up to mg/L as reported by Caux et al. (Caux PY Kent RA Fan GT Grande C (1998) Canadian water guidelines for linuron. Environ Toxic Water13: 1–41). In this article we did not perform the consumer risk analysis but we investigated the effects of these 2 chemicals on the nervous system, in particular looking at the molecular action.
Reviewer 2 Report
The paper entitled " An experimental approach to study the effects of Realistic Environmental Mixture of Linuron and Propamocarb on zebrafish 3 synaptogenesis” is interesting and suitable to publish in the journal. Based on the quality of paper and analysis, I recommend a minor revision.
My main concerns are:
- A nexus of pesticide and human exposure has been investigated in previous study as well. I suggest authors to review and support the statements in introduction with the given studies
[1] Use of artificial neural networks to rescue agrochemical-based health hazards: A resource optimisation method for cleaner crop production. Journal of Cleaner Production. Volume 238, 20 November 2019, 117900. Doi. 10.1016/j.jclepro.2019.117900
[2] Fairtrade, Agrochemical Input Use, and Effects on Human Health and the Environment.
- I highly suggest to authors to add a paragraph of literature review (about 300 words) in introduction to explore study gape, novelty and then objectives of the study. The literature review must be related to topic of the study.
- A structure of article should be added at the end of introduction for instance Section 1 belongs to the study background, section 2 relates to research methods etc….
- There should be a proper numbering of heading in section of methodology for instance 2.1, 2.2, 2.3…..
- A limitation of study should be added at the end of introduction and recommendation for future study.
Author Response
We have modified and highligthed in yellow the changes that we have done following the Reviewer suggestions.
- A nexus of pesticide and human exposure has been investigated in previous study as well. I suggest authors to review and support the statements in introduction with the given studies
[1] Use of artificial neural networks to rescue agrochemical-based health hazards: A resource optimisation method for cleaner crop production. Journal of Cleaner Production. Volume 238, 20 November 2019, 117900. Doi. 10.1016/j.jclepro.2019.117900
[2] Fairtrade, Agrochemical Input Use, and Effects on Human Health and the Environment.
As suggested by the Reviewer we have added these 2 references
- I highly suggest to authors to add a paragraph of literature review (about 300 words) in introduction to explore study gape, novelty and then objectives of the study. The literature review must be related to topic of the study.
We added a brief paragraph of the literature review in the introduction, as suggested by the Reviewer.
- A structure of article should be added at the end of introduction for instance Section 1 belongs to the study background, section 2 relates to research methods etc….
We modified the introduction, following the Reviewer’s suggestion, without adding the section because the journal instruction does not expect it. We use paragraphs to distinguish the different sections. We hope that now it appears more consequential and well organized.
- There should be a proper numbering of heading in section of methodology for instance 2.1, 2.2, 2.3…..
Done
- A limitation of study should be added at the end of introduction and recommendation for future study.
Done